# Design of Etched- and Functionalized-Halloysite/Meloxicam Hybrids: A Tool for Enhancing Drug Solubility and Dissolution Rate

**DOI:** 10.3390/pharmaceutics16030338

**Published:** 2024-02-28

**Authors:** Valeria Friuli, Claudia Urru, Chiara Ferrara, Debora Maria Conti, Giovanna Bruni, Lauretta Maggi, Doretta Capsoni

**Affiliations:** 1Department of Drug Sciences, University of Pavia, Via Taramelli 12, 27100 Pavia, Italy; 2Department of Chemistry, Physical Chemistry Section & CSGI (Consorzio Interuniversitario per lo Sviluppo dei Sistemi a Grande Interfase), University of Pavia, 27100 Pavia, Italy; claudia.urru01@universitadipavia.it (C.U.); deboramaria.conti01@universitadipavia.it (D.M.C.); giovanna.bruni@unipv.it (G.B.); doretta.capsoni@unipv.it (D.C.); 3Department of Materials Science, University of Milano-Bicocca, Via Cozzi 55, 20125 Milano, Italy; chiara.ferrara@unimib.it

**Keywords:** halloysite nanotubes, meloxicam, dissolution tests, drug–nanoclay composites, poorly soluble drugs

## Abstract

The study focuses on the synthesis and characterization of Meloxicam–halloysite nanotube (HNT) composites as a viable approach to enhance the solubility and dissolution rate of meloxicam, a poorly water-soluble drug (BCS class II). Meloxicam is loaded on commercial and modified halloysite (acidic and alkaline etching, or APTES and chitosan functionalization) via a solution method. Several techniques (XRPD, FT-IR, ^13^C solid-state NMR, SEM, EDS, TEM, DSC, TGA) are applied to characterize both HNTs and meloxicam–HNT systems. In all the investigated drug–clay hybrids, a high meloxicam loading of about 40 wt% is detected. The halloysite modification processes and the drug loading do not alter the structure and morphology of both meloxicam and halloysite nanotubes, which are in intimate contact in the composites. Weak drug–clay and drug-functionalizing agent interactions occur, involving the meloxicam amidic functional group. All the meloxicam–halloysite composites exhibit enhanced dissolution rates, as compared to meloxicam. The meloxicam–halloysite composite, functionalized with chitosan, showed the best performance both in water and in buffer at pH 7.5. The drug is completely released in 4–5 h in water and in less than 1 h in phosphate buffer. Notably, an equilibrium solubility of 13.7 ± 4.2 mg/L in distilled water at 21 °C is detected, and wettability dramatically increases, compared to the raw meloxicam. These promising results can be explained by the chitosan grafting on the outer surface of halloysite nanotubes, which provides increased specific surface area (100 m^2^/g) disposable for drug adsorption/desorption.

## 1. Introduction

Among drugs properties, solubility in water represents a basic and crucial parameter as it affects the active pharmaceutical ingredient (API) efficacy [1]. Solubility has an impact on drug bioavailability. It is established that about 40% of APIs currently marketed are poor water-soluble drugs, with solubility values lower than 0.1 mg/mL (United States Pharmacopeia (USP) definition) [2], and belong to classes II and IV of the biopharmaceutics classification system (BCS). In particular, class II drugs are poorly soluble in both water and conventional organic solvents. These drugs are barely absorbed and have limited therapeutic efficiency and biosafety, which affect API potency. Both the pharmaceutical industry and research institutions have devoted time and invested resources to develop suitable approaches to enhance drug bioavailability and water solubility. The traditional methodologies [1,3,4] include particle size reduction with specific surface area increase, use of surfactant and co-solvents [5,6], stabilization of amorphous and new crystal phases [7], formation of salts [8], co-crystals [9,10,11], complexes [12], and inclusion compounds [13,14].

The recent fast progress of nanoscience and nanotechnology paves the way for the development of new organic and inorganic nanomaterials as vehicles of poorly soluble drugs [15,16]. These new approaches take advantage of the appealing physicochemical properties of the nanocarriers, such as small particle size, high surface area, porosity, variable morphology, and adsorption capacity, to load/bind drugs and enhance their solubility and bioavailability. Among nanocarriers, particular attention is focused on clay mineral materials, both natural and synthetic [17]: they display different morphologies (nanoplates, nanotubes, nanofibers), and controllable size and porosity. Improved solubility and dissolution rates have been obtained for a variety of drugs loaded into unchanged and modified clays such as mesoporous silica, laponite, sepiolite, montmorillonite, and halloysite [18,19,20,21].

Halloysite (Al_2_Si_2_O_5_(OH)_4_·nH_2_O, *n* = 0–2) is an aluminosilicate clay of the kaolin group. It is a naturally widespread mineral with remarkable properties, ubiquitous in soils of wet climates, and weathered rocks. The di-hydrate form, known as halloysite-10 Å, easily releases the weakly held water molecules, giving rise to the more common anhydrous phase, referred to as halloysite-7 Å [22,23,24]. Halloysite consists of a layered structure, based on Si-O tetrahedra and Al-O octahedra sheets connected via oxygen to form a single layer. The halloysite nanoparticles adopt a variety of morphologies, from platelet-like to spheroidal. Still, the most common form is the nanotubular one, as a result of the aluminosilicate layer wrapping induced by the dimensional mismatch of the silica and alumina polyhedra [25]. The Halloysite nanotubes (HNTs) properties are strictly related to both morphology and chemical features. They display a high length/diameter aspect ratio (length: 0.4–1 mm, outer diameter 20–200 nm), a lumen (diameter: 10–70 nm), specific surface areas of about 50–60 m^2^/g, and pore size distribution in the 2–50 nm range [26]. The siloxane and silanol groups on the outer surface and the aluminol on the inner one infer a negatively and positively charged nature to the two surfaces, respectively [27]. All these features, combined with the cheapness of halloysite (large number of deposits), non-toxicity, and biocompatibility [28,29], make HNTs suitable for a variety of applications in catalysis, polluted water remediation and environmental depuration, drug carrier and delivery systems, and polymer fillers [30]. The large number of applications can be further widened by HNT modifications, which aim to make the internal and external surfaces of halloysite nanotubes more reactive and selective and to make the interlayer space accessible to specific molecules. The most common modification strategies employ acidic and alkaline etching, functionalization, thermal treatments, and intercalation [31,32,33]. Among HNT surface modification strategies, the functionalization with organosilane demonstrates a performant approach to control the loading and profile release of drugs. The silanol groups of organosilanes interact with the OH groups of the halloysite surfaces, mainly with the aluminol present in the inner HNT surface [34]. The grafted organosilanes provide new functionalities suitable to modify the drug loading and release [35,36]. As concerns the outer HNT surface modification, functionalization with cationic polymers, such as chitosan (CTS), is widely used. CTS, obtained via partial deacetylation of chitin, is a copolymer of N-acetylglucosamine and glucosamine. It is low-cost, widely available, and rich in -OH and -NH_2_ functional groups, suitable to give hydrophilic properties and reactive lone pair electrons, respectively [37]. In acidic conditions, an electrostatic interaction occurs between the protonated amine group and the negatively charged external surface of the halloysite [38,39].

Meloxicam (C_14_H_13_N_3_O_4_S_2_) is an enolic acid oxicam derivative with the molecular structure shown in Figure 1. It is a non-steroidal anti-inflammatory drug (NSAID) applied in therapy for the long-term treatment of musculo-skeletal complaints, such as rheumatoid arthritis, osteoarthritis, and other joint diseases [40]. It also displays anticancer effects in some human tumor cells [41]. As concerns crystal structure, five forms are reported in the literature [42,43]. Form I, II, III, and V are anhydrous, but only the crystal structure of Form I is completely addressed (triclinic). Form I is suitable for commercialization [44]. Form IV is hydrated and displays an orthorhombic structure.

Meloxicam is a class II drug (low solubility and high permeability): it is practically insoluble in water (and weakly soluble in common organic solvents such as methanol and ethanol). This implies a poor dissolution rate and variable bioavailability after oral administration. Various approaches were applied to enhance its solubility and to improve its dissolution, including particle size reduction, formation of co-crystals and drug-cyclodextrin inclusion compounds, drug insertion into double-layered hydroxides, or taking advantage of spray drying techniques [45,46,47].

In this work, we synthesize meloxicam–halloysite and meloxicam-modified halloysite composites, and we investigate the strategy’s effectiveness in enhancing the drug solubility and dissolution rate. To the best of our knowledge, these composites have not yet been investigated. Different approaches were applied to modify the HNTs before meloxicam loading: acid and alkaline etching and functionalization with APTES and CTS, to remodel the inner and outer surface, respectively. Both modified HNTs and drug–halloysite hybrid systems were thoroughly characterized using a variety of techniques (XRPD, FT-IR, SEM, EDS, TEM, thermal analysis, and specific surface area evaluation) to demonstrate the effective drug loading and evaluate the morphology of the composites and drug–halloysite interactions. The pharmaceutical characterization concerned the drug loading, the wettability (contact angle), the solubility, and the in vitro dissolution rate in different biorelevant fluids: deionized water, 0.1 N HCl at pH 1 (which simulates the gastric environment In fasted state), phosphate buffer at pH 4.5 (which simulates fed state), and pH 7.5 buffer as requested by the US Pharmacopoeia [48], at 37 °C.

## 2. Materials and Methods

All the chemicals employed were technical grade or higher in quality. Meloxicam (MEL, C_14_H_13_N_3_O_4_S_2_) was kindly given by AMSA S.p.A. (Milano, Italy). Halloysite nanotubes (HNTs, Al_2_Si_2_O_5_(OH)_4_·2H_2_O, 685445), hydrochloric acid (HCl, 1.37007), sodium hydroxide (NaOH, 1.06462), methanol (MeOH, 179337), ethanol (EtOH, 1.00986), acetic acid (glacial, ≥99.7%, 695092), oleic acid (tech., 90%, 364525), chitosan (CTS, (C_8_H_13_NO_5_)_n_ low m.w., deacetylation ≥75%, 448869), and (3-aminopropyl)triethoxysilane (APTES, C_9_H_23_NO_3_Si, 99%, 440140) were purchased from Merck (Milano, Italy).

### 2.1. Synthesis

#### 2.1.1. Acid Etching of HNTs

The acid activation of HNTs was performed with HCl, following the procedure suggested by Wang et al. [49] with some modifications. Firstly, 1 g of commercial HNTs was suspended in 10 mL of HCl solution (2 and 4 mol L^−1^) at room temperature. The obtained suspension was centrifuged at 6000 rpm for 5 min after 4 h stirring. The precipitate was washed with distilled water until pH ≈ 6 in the supernatant was reached. The products were dried overnight at 105 °C and then ground in an agate mortar.

#### 2.1.2. Alkaline Etching of HNTs

The alkaline activation of HNTs was performed with NaOH by following the procedure suggested by Wang et al. [50] with some modifications. Firstly, 1 g of commercial HNTs was dispersed in 10 mL of NaOH solution (0.5 mol L^−1^). The mixture was centrifuged at 6000 rpm for 5 min, after 1 h of treatment in ultrasonic bath. The precipitate was washed with distilled water until pH ≈ 7 in the supernatant was reached. The products were dried overnight at 105 °C and then ground in an agate mortar.

#### 2.1.3. HNTs Functionalization with APTES

The functionalization of HNTs with organosilane was performed by following the procedure reported by Krishnaiah et al. [34]. Firstly, 1 mL of APTES was dissolved in 40 mL of EtOH. Some drops of acetic acid were added to adjust the pH between 4.5 and 5.5. The solution was magnetically stirred at 60 °C for 15 min, then 4 g of HNTs were added, keeping stirring and heating for 2 h. The product was centrifuged and washed several times with EtOH. The sample was dried at room temperature for 24 h and then dried under vacuum at 100 °C for 8 h to remove moisture and ethanol.

#### 2.1.4. HNTs Functionalization with Chitosan

The functionalization of HNTs with chitosan was performed by following the procedure reported by Li et al. [39]. A suspension of HNTs in distilled water (50 mL, 1 wt%) was sonicated for 1 h. Meanwhile, chitosan (0.7 g) was dissolved in a glacial acid acetic aqueous solution (50 mL, 1 wt%). The HNTs and the chitosan solutions were mixed under continuous stirring for 12 h to reach the adsorption equilibrium. The solution was then emulsified with oleic acid (25 mL) and aged for 12 h. EtOH was then added to precipitate the microspheres. The product was finally centrifuged (6000 rpm, 5 min) and washed with distilled water several times.

#### 2.1.5. Meloxicam Loading onto HNTs

The loading of the active principle onto commercial, etched, and functionalized HNT samples was performed by following the protocol reported by Neolaka et al. [51]. The procedure was performed at room temperature. Firstly, 130 mg of meloxicam were dissolved in 15 mL of MeOH, then 200 mg of HNTs were added (theoretical loading: 40 wt%). The solution was magnetically stirred for 24 h and then centrifuged for 5 min at 6000 rpm. The solid was dried at 35 °C for 2 h.

### 2.2. Characterization Techniques

X-Ray Powder Diffraction (XRPD) measurements were carried out using a Bruker D5005 diffractometer (Bruker, Karlsruhe, Germany) equipped with a curved-graphite monochromator, scintillation detector, and CuKα anticathode as a source. The data collection was performed in the angular range 7° ≤ 2θ ≤ 48° with step size of 0.03° and counting time of 5 s/step, on powders set into a silicon zero-background sample holder.

Fourier Transform Infrared spectroscopy (FT-IR) spectra were collected with a Nicolet FT-IR iS10 Spectrometer (Nicolet, Madison, WI, USA) equipped with an ATR (Attenuated Total Reflectance) sampling accessory (Smart iTR with diamond plate) by co-adding 32 scans in the 4000–650 cm^−1^ range at 4 cm^−1^ resolution.

^13^C data were acquired using the ^1^H-^13^C cross polarization (CPMAS) pulse sequence under 10 kHz spinning speed MAS condition. The ^1^H 90 deg. pulse was set to 2.5 ms, and the delay time 5–20 s (calibrated for each sample). The spectra have been acquired using 1 k scans and referenced to the adamantane signals used as secondary standard.

Scanning Electron Microscopy (SEM) images have been acquired using a Zeiss EVO MA10 microscope (Carl Zeiss, Oberkochen, Germany) on gold-sputtered samples in argon atmosphere. The microscope is equipped with an energy dispersive detector (X-max 50 mm^2^, Oxford Instruments, Oxford, UK) for the energy dispersive X-ray spectroscopy (EDS) analysis.

Transmission Electron Microscopy (TEM) images were collected with a JEOL JEM-1200 EX II microscope (JEOL Ltd., Tokyo, Japan) equipped with a TEM CCD camera Olympus Mega View III (Olympus soft imaging solutions (OSIS) GmbH, from 2015 EMSIS GmbH, Munster, Germany). Micrographs were obtained at 100 kV high voltage (tungsten filament gun) and 100 kX magnification with 1376 × 1032 pixel format. Samples were prepared by drop-casting the suspension on formvar/carbon-coated nickel grids.

The specific surface area of the materials was evaluated via N_2_ adsorption in a Sorptomatic 1990 porosimeter (Thermo Electron, Waltham, MA, USA); the Brunauer-Emmett-Teller (BET) method was applied. Samples were degassed at 25 °C overnight before the measurement.

The thermal characterization was performed by using a DSC Q2000 apparatus and a simultaneous TGA/DSC SDT600 apparatus (ultrapure indium is used to calibrate temperature and enthalpy), both interfaced with a TA 5000 data station (TA Instruments, NewCastle, DE, USA). Samples were scanned from room temperature to 300 °C at 10 K·min^−1^ in open standard aluminum pans under a continuously purged dry nitrogen atmosphere (3 L·h^−1^).

### 2.3. Drug Content

The meloxicam content of the samples was determined in phosphate buffer at pH 7.5 in which the active ingredient has high solubility. MEL pka = 4.08 [52]. The solutions were analyzed with a UV spectrophotometer (Lambda 25, Perkin-Elmer, Monza, Italy) using a calibration curve obtained in the same conditions (correlation coefficient: 0.9999). The drug loadings were used to define the weight of the samples to be subjected to the dissolution test and corresponding to a MEL dose of 7.5 mg. The UV spectrum of halloysite was previously recorded to verify that it did not interfere with the MEL absorbance at the wavelength of 362 nm.

### 2.4. Dissolution Test and Solubility

The dissolution tests were performed on pure MEL and then on commercial-, modified, and functionalized MHNT compounds. All the samples were previously sieved through a 230-mesh grid, 63 μm, (Endecotts, London, UK). The USP apparatus 2 paddle (Erweka DT-D6, Erweka GmbH, Dusseldorf, Germany) at 37.0 ± 0.5 °C (three replicates) was used.

The samples, always containing 7.5 mg of MEL, were tested at 75 rpm in 900 mL of four different fluids: pH 7.5 phosphate buffer, according to the official monograph reported in the US Pharmacopoeia [48], HCl 0.1 N, pH 1.0 (which simulates the gastric environment in fasted conditions), pH 4.5 phosphate buffer (which simulates the gastric environment in fed conditions), and deionized water (which simulates the assumption with a glass of water). All the dissolution media were prepared according to the “Reagent and buffer solutions section” of the USP [53]. MEL concentrations were measured using UV-Vis absorbance (Lambda 25, Perkin-Elmer, and Monza, Italy) on filtered portions of the dissolution medium at 362 nm. The data were processed using suitable software (Winlab V6 software, Perkin-Elmer, Monza, Italy) to obtain the dissolution profiles.

The dissolution tests were carried out in two fluids simulating the gastric environment (in fed and fasted conditions) because the drug first encounters these conditions in vivo, and also because the drug has a pH-dependent solubility, less soluble in acidic and neutral environments, and more soluble in a basic environment; thus, these conditions are the most selective to highlight the differences in the behavior of the proposed compounds. The USP test was used anyway because it is an essential reference for all products containing oral meloxicam. At the same time, deionized water is very useful for understanding the variability of the behavior of the hybrids under examination in unbuffered conditions, when the very presence of the MEL (and possibly the carrier) could also significantly modify the pH of the medium, as detected during the dissolution test.

Deionized water was used to have a reference in an unbuffered fluid. Administration with a glass of water could dilute the buffered conditions of the gastric environment, modifying the pH conditions.

The shake-flask method was used to measure the solubility of MEL, MH_C, and MH_A in deionized water at 21 °C [54]. An aliquot of the supernatant was withdrawn from the flask at predetermined time intervals and filtered through 0.45 μm Millipore filters. The amount of drug solubilized was determined through UV detection. The analysis was repeated until equilibrium was reached (three replicates).

### 2.5. Contact Angle

Wettability was determined via contact angle measurements. Different images were taken of the interaction during the time, of a 9 µL drop (of the same fluids used for the dissolution test) deposited on the surface of the different samples in powder. Contact Angle Meter DMe-211Plus (NTG Nuova Tecnogalenica, Cernusco, Italy) acquires the images of the drop at progressive times (from t = 0 up to 300 s) and suitable software elaborates the results in terms of the angle between the liquid–vapor interface and liquid–solid interface. Three replicates were performed for each sample.

## 3. Results and Discussion

Commercial HNTs were modified via acidic or alkaline etching or via functionalization (CTS and APTES). All treated samples were characterized and compared to raw halloysite. The solution method was employed for loading meloxicam on both commercial and modified HNTs. A list of the synthesized samples and related alphanumeric codes is reported in Table 1.

The dissolution tests were carried out on meloxicam and all the drug–halloysite systems (see Section 3.1.8). In the case of drug-etched halloysite composites, we did not obtain impressive improvements. Here, we report the characterization of the alkaline-etched sample, which gives moderate enhancement in the dissolution profiles at low pH; the results of the acidic-etched samples are reported in Appendix A.

### 3.1. Samples Characterization

#### 3.1.1. XRPD

Figure 1 shows XRPD patterns of (*a*) meloxicam, (*b*) commercial HNTs, and (*c*–*f*) selected composites. Figure 1 (*a*) displays a pattern typical of a well-crystallized powder, ascribable to meloxicam enolic form I [42,44]. The main diffraction peaks are detected at 2θ values: 11.3, 13.1, 13.5, 15.0, 16.0, 16.7, 17.9, 18.7, 19.3, 19.7, 20.5, 22.0, 23.5, 24.0, 25.9, 26.4, 27.0, 28.0, 28.6, 29.4, 29.6, 32.0, 36.0, 39.9, 40.7, 41.4, 42.1, and 43.8°. The halloysite pattern (Figure 1 (*b*)) compares well with the literature ones [55], and peak positions match the data reported in the JCPDS database for halloysite (PDF# 029-1487). The reflection at about 12° corresponds to the (0 0 1) d-spacing of 7.35 Å. It is detected in the anhydrous halloysite form (halloysite-7 Å), and easily obtained by the di-hydrated one (halloysite-10 Å) near room temperature [22,24]. The peak at 24° is attributed to the (0 0 2) reflection (*d* = 3.63 Å). The absence of the reflection at 8.8°, typical of halloysite-10 Å, further confirms that there is no interlayer water. [24] The broad peak at 20° is assigned to the (1 0 0) reflection and demonstrates that the halloysite has a nanotubular morphology [56]. The peaks at about 35° and 37.9° are associated with (1 1 0) and (0 0 3) reflections, respectively. Small amounts of kaolinite 1 A and quartz impurities, usually present in halloysite clays, are detected (sharp peaks at about 10.1° and 26.6°) [57]. Modified HNT samples do not show changes in crystalline or nanotubular structure apart from the type of treatment (Appendix A). This is consistent with the literature data: the HCl etching [49] and the NaOH one at mild alkaline concentrations (up to NaOH 4 M) [50] do not affect the crystalline nature of the halloysite. In the functionalized halloysites, the presence of the peak at about 12°, typical of Halloysite-7 Å, confirms that both chitosan and APTES moieties are not intercalated into the interlayer space and may interact only with the nanotube surfaces [38,58,59]. Drug–clay patterns (Figure 1 (*c–f*) and Appendix A) show the presence of both halloysite and meloxicam reflections, suggesting a successful loading and demonstrating that neither crystalline structure changes nor decomposition occurs in both components. The meloxicam crystallite size in original and loaded samples is evaluated by applying the Scherrer equation to the peaks at about 15° and 26° (intense and poorly overlapped). The results are reported in Appendix A. The crystallite size of meloxicam is smaller when loaded onto modified halloysite, compared to the MEL and MH samples. The results suggest that surface modifications (both etching and functionalization) may affect the meloxicam crystallization by reducing the crystallite size. Finally, the absence of the reflection peak at 8.8° also in loaded samples suggests that the drug did not intercalate into the interlayer space of the HNTs. The XRPD results are confirmed also by the ^13^C NMR spectra (Appendix A) collected for the MEL, MH, MH-A, and MH-C samples. The obtained spectra reveal that the meloxicam structure is not modified, and the peaks are not shifted or broadened, suggesting that the crystal structure of the compound is maintained and thus it is possible to infer the absence of intercalation into the halloysite layers (that, on the contrary, would involve strong interactions and thus changes in the observed chemical shifts). Combining the results from XRPD and NMR analysis, it is thus possible to suggest that both the halloysite and meloxicam structure are not strongly modified in the MH, MH-A, and MH-C samples with respect to the MEL and H original compounds.

#### 3.1.2. FT-IR Spectroscopy

Figure 2I,II shows unmodified and functionalized halloysite, and functionalizing agent spectra. Positions and assignments of the characteristic bands of meloxicam, halloysite, APTES, and chitosan are reported in Appendix A. The commercial HNT spectrum (Figure 2I,II (*a*)) compares well with those reported in the literature [60]. The very weak bands at about 3545 cm^−1^ and 1647 cm^−1^ suggest the presence of traces of interlayer water molecules, not detectable with the XRPD technique [56]. The APTES spectrum (Figure 2I,II (*b*)) matches the literature data [61]. The most relevant signals, detectable also in the H_A spectrum (Figure 2I,II (*c*)) are those associated with C-H stretching of APTES backbone and ethoxy groups, in the range 2800–3000 cm^−1^, and those attributed to NH_2_ and NH_3_^+^ bending at 1440 and 1482 cm^−1^, respectively [62,63]. After APTES functionalization, halloysite bands related to Al-OH stretching modes shift to 3687 and 3617 cm^−1^, suggesting an interaction between APTES and internal aluminol groups. These interactions are confirmed by the shift of C-H signals of APTES in H_A (see Table 2), and by a decrease in intensity of the signal related to the Al-OH of inner surface groups. This indicates that the grafting took place between Al-OH groups and hydrolyzed APTES, and mainly involves the inner surface of HNTs: however, possible APTES grafting on the external surface, which exhibits reactive hydroxyl groups as surface defects, cannot be ruled out [64]. The chitosan spectrum (Figure 2I,II (*d*)) compares well with those reported in the literature [65,66]. Signals related to amidic stretching are detected ($\bar{\nu }$ = 1651, 1560, and 1316 cm^−1^) because a certain amount of chitin is present (deacetylation degree ≥ 75%). In the H_C spectrum, the main bands of halloysite and chitosan are detected, suggesting the successful functionalization of the nanoclay. A shift of the chitosan signals related to the symmetric and asymmetric C-H stretching and to the amidic C=O stretching is detected in the H_C sample (Table 2). In particular, the peak at 1707 cm^−1^ confirms the hydrogen bonding between CTS and -OH groups on the halloysite, and the electrostatic interaction between CTS polycations and the negative-charged outer surface of the halloysite [67]. Notably, no changes are detected for the Al-OH inner surface hydroxyl groups (3600–3700 cm^−1^ range). These evidences confirm the chitosan grafting of the outer surface of halloysite nanotubes. The FT-IR spectra of the etched halloysite are shown in Appendix A. Both acidic and alkaline etching treatments do not modify the characteristic absorption bands of the clay or generate undesired subproducts. The FT-IR results are consistent with the literature data [49,50], and with XRPD, confirming the nanoclay structure is preserved after etching.

Meloxicam FT-IR spectrum (Figure 3 (*a*)) agrees with those reported in the literature for the polymorphic form I [42,43]. This result confirms the XRPD one. The remarkable band at 3285 cm^−1^ is attributed to the N-H stretching and it is peculiar to the meloxicam polymorph I [68,69,70]. The amidic C=O stretching signal is detected at 1616 cm^−1^, and the aromatic ones are located at 1548, 1524, 1456, and 1446 cm^−1^. The bands at 1344 and 1182 cm^−1^ are attributed to the SO_2_ asymmetric and symmetric stretching, respectively [46] (see Appendix A).

Drug–clay systems spectra (Figure 3I,II (*b–e*), and Appendix A) show the simultaneous presence of both halloysite and meloxicam signals, supporting the hypothesis of a successful loading. In addition, the bands peculiar to the functionalizing agents are detected in the drug-functionalized halloysite hybrids. A shift of the amidic N-H band of the meloxicam is detected in functionalized and commercial halloysite–drug systems (see Table 2). This suggests that drug–clay interaction occurs via the amidic group. The small shift of C-H stretching bands of APTES and CTS in drug–clay samples compared to the functionalized halloysite ones suggests possible weak interactions between meloxicam and the functionalizing agents.

The FT-IR results confirm the successful functionalization of the halloysite nanotubes, which mainly occurs on the inner surface for APTES, and on the outer one for CTS. In all drug–halloysite and drug-modified halloysite samples, meloxicam is successfully loaded onto HNTs. Based on the bands shifts, weak interactions occur between the drug and the support in MH and in the functionalized systems. No significant changes are detected in the etched systems.

#### 3.1.3. SEM and EDS

Figure 4a shows the SEM image of the unmodified HNTs. Aggregates of variable size (2–20 µm) composed of nanotubular sub-particles are observed. Alkaline etching (Figure 4b) does not modify halloysite morphology. In the case of acidic etching (Appendix A), the nanotubular sub-structure is preserved, but nanotubes are glued to form aggregates with compact surfaces. APTES and CTS functionalization (Figure 4c and Figure 4d, respectively) do not infect the tubular structure but, especially in the case of CTS functionalization, also aggregates of rounded particles are detected, possibly due to biopolymer ability to connect nanotubes and induce hierarchical and interconnected pore distribution [39].

Meloxicam is composed of micrometric prismatic crystals with a smooth surface (Figure 5). In drug–clay systems, both meloxicam particles and halloysite nanotubes are observed, the latter covering the drug surface (Figure 6a–d and Appendix A), without changes in the morphology of single components. A detailed investigation at higher magnification (Appendix A) evidences that the HNTs form aggregates on the meloxicam surface for MH_HCl_2M, MH_HCl_4M, and MH_A samples. In the case of MH, MH_NaOH_0.5M, and MH_C, HNTs are less clusterized and, especially for the chitosan functionalization, penetrate the drug surface giving rise to a more intimate contact between the drug and the clay.

EDS analysis is applied to evaluate the distribution of the elements in drug–clay systems. S is chosen to identify meloxicam, Al, and Si to investigate halloysite. The distribution maps are shown in Appendix A. As expected, the Si and Al distribution for each drug–clay system is comparable. Also, S distribution fairly compares to the Al and Si ones; these evidences support the SEM results showing nanotubes aggregates on meloxicam particles. Based on the S, Al, and Si content detected using EDS microanalysis, the meloxicam amount in each drug–clay system is evaluated. The results are reported in Appendix A and are in fair agreement with the synthesis composition (40 wt%).

#### 3.1.4. TEM

Figure 7 and Appendix A show TEM images of raw and modified halloysite. Figure 7a confirms the nanotubular structure of commercial halloysite. Nanotubes are characterized by an external diameter ranging from 50 to 80 nm, and by an internal lumen averagely large 15 nm. Alkaline etching (Figure 7b) tends to thin nanotube walls and to open the lumen, while acidic conditions make the nanotube surface rough (Appendix A) [49]. This evidence is supported by external/internal diameter ratio evaluated on several nanotubes for H, HCl-, and NaOH-etched samples. The ratio is higher than 3.3 for H and acidic etching, and lower than 3 for alkaline one. In addition, alkaline etching disaggregates nanotubes in suspension [50]. Functionalization (Figure 7c,d) does not change the nanotubular nature of HNTs, but the chitosan one (Figure 7d) displays a rough and uneven external surface. This evidence suggests the chitosan is grafted on the outer surface of the halloysite nanotubes, as reported in the literature [71].

In Figure 8 the TEM images of meloxicam and drug–clay samples are shown. In the MEL sample (Figure 8a), one of the smaller particles can be observed, with a prismatic shape consistent with the SEM observations (Figure 5). This kind of particles can also be distinguished in all composite samples (Figure 8b–e and Appendix A) and they are in intimate contact with halloysite nanotubes confirming once again the drug–clay composite formation.

#### 3.1.5. BET Analysis

Table 3 shows the values of the specific surface area of raw and modified halloysite. The result obtained for unmodified HNTs compares to the literature values [26]. Functionalization leads to a decrease in the specific surface area for H_A: this evidence is consistent with the possible grafting of APTES on the inner surface of halloysite nanotubes, as also suggested by changes in the FT-IR bands attributed to Al-OH groups of the inner surface of halloysite nanotubes [72]. In the case of H_NaOH_0.5M and H_C samples, an increase is registered [39,73], more pronounced for the chitosan functionalized halloysite. The H_C sample does not provide only a larger support surface area but also connects nanotubes possibly forming interconnected pore distribution, beneficial for drug adsorption/interaction [39].

#### 3.1.6. DSC

DSC curves of raw meloxicam and halloysite and of the drug–clay composites are shown in Figure 9. In the H sample only a small endothermic event at T_onset_ = 117 °C (see Figure 9 inset), due to a dehydration process, is detected [74]. This evidence confirms the FT-IR results suggesting the presence of traces of interlayer water molecules, not detectable with the XRPD technique. The DSC curve of the MEL sample (Figure 9 (*e*)) compares well with the literature data [43,75]: it shows an endothermic peak at T_onset_ = 265 °C (ΔH = −158.5 J/g), attributed to melting/decomposition, as confirmed by the mass loss detected at approximately the same temperature in the TGA curve (Appendix A). The peak is followed by a second endothermic peak, which is more irregular due to further decomposition. Functionalizations do not introduce or modify thermal events in halloysite samples. In all the drug–clay samples the thermal events related to the meloxicam melting/decomposition shift to lower temperature with respect to the M sample, as reported in Table 4.

TGA curves of the original meloxicam and the MH and MH_C samples, taken as examples, are shown in Appendix A. The data confirm that also the decomposition process occurs at lower temperatures in drug–clay composites compared to meloxicam. This behavior may be explained by the weak drug–halloysite interaction evidenced by FT-IR analysis and/or by the smaller crystallite size of meloxicam when loaded on modified halloysite, as evaluated using XRPD data.

#### 3.1.7. Drug Content

The drug loading determined using UV-Vis spectroscopy was very high for all samples (Table 5), and this factor is extremely important for the further development of these products for oral administration. The obtained amount compares to that evaluated using EDS microanalysis (Appendix A).

#### 3.1.8. Dissolution Rate and Solubility

As expected, all the proposed samples show a much lower drug release rate in acidic fluids (pH 1.0 and pH 4.5) where the drug is less soluble, but always faster than M alone (Figure 10 and Figure 11).

In water, this difference is greatly expanded, and it is therefore able to characterize the behavior of the different products proposed. MH_C gives the best result because it can release the entire dose of the drug in approximately 4–5 h. Even more indicative is the behavior of the samples in pH 7.5 buffer (a condition in which the drug is more soluble). Indeed, in this medium, MH_A, and especially MH_C, release the drug much faster than MEL, and also than MH. The drug release is completed in 1.5 h, or less than 1 h, respectively. The solubility of MEL is highly pH dependent, in fact in unbuffered water the dose of the drug lowers the pH of the medium from 6.9 to 6.3, a value measured at the end of the dissolution test. While in other buffered fluids, the final pH does not change.

As highlighted by the contact angle measurements (see below), wettability plays an important role in the dissolution phase of MEL and this factor could explain the high SD of some of the dissolution profiles of the analyzed compounds, in particular, those characterized by an intermediate speed (not too fast and not too slow), such as those recorded water and in phosphate buffer at pH 7.5. In these fluids, the powdered product initially remains on the fluid surface, then progressively gets wet, and then slowly sinks into the dissolution medium. In the other fluids, it tends to remain on the surface for the entire duration of the test.

So, the solubility was measured on these two products. Indeed, the results confirmed those of the dissolution tests. The equilibrium solubility, measured in deionized water at 21 °C, was: 8.0 ± 0.2 mg/L for MEL; only a little higher for MH_A: 9.0 ± 1.0 mg/L and much higher for MH_C: 13.7 ± 4.2 mg/L.

The obtained dissolution results can be related to the chemical–physical properties of the modified halloysites and of the drug–clay systems. The etching process (both alkaline and acid) poorly alters the halloysite nanotubes morphology and no relevant drug–clay interaction is evidenced by FT-IR analysis. In the case of MH, MH_A, and MH_C samples, FT-IR spectra demonstrate weak drug–clay interactions. For the CTS-functionalized drug–clay composite the CTS itself, grafted on the outer surface of HNTs, weakly interacts with meloxicam and increases the specific surface area of the carrier. These features play a relevant role in the adsorption/desorption process and are beneficial to improving the meloxicam dissolution rate and solubility in water.

#### 3.1.9. Wettability

The MH_C sample was therefore the most promising and suitable for further development. To better explain the causes of its improved behavior, this compound was also subjected to the contact angle test in the four fluids considered (Figure 12 and Appendix A). Contact angle measurements demonstrate how hydrophobic and non-wettable the drug is, while loading onto the functionalized HNT dramatically increases the wettability of the compound.

## 4. Conclusions

In this work, new meloxicam–halloysite and meloxicam-modified halloysite hybrids are synthesized via an easy and feasible solution approach. The results obtained by several characterization techniques on the drug–clay systems demonstrate that, independently of the applied modification (alkaline/acidic etching, or APTES and chitosan functionalization), (i) a meloxicam loading as high as 40 wt% is reached, (ii) the meloxicam and halloysite are in intimate contact, (iii) their structures and morphologies are retained, (iv) smaller crystallite size of meloxicam is detected when loaded on modified halloysites. FT-IR results evidence that weak drug-carrier and drug-functionalizing agents interactions occur, as suggested by the shift of the meloxicam amidic N-H band and of the C-H stretching bands of APTES and CTS in drug–clay systems compared to the unloaded samples. Improved dissolution rates are obtained in the drug-loaded hybrids compared to the raw meloxicam, more relevant in water and at pH 7.5. At these pH values, the meloxicam–CTS modified halloysite is the most performant: the chitosan, grafted on the external surface of the halloysite nanotubes, increases the nanotubes surface area and weakly interacts with meloxicam. These features affect the drug adsorption/desorption process and impact the dissolution profile and the equilibrium solubility. Also, the better wettability can explain the faster dissolution rate of the system. The preliminary evaluation of the possible modifications of the halloysite carrier to improve the dissolution performance of MEL allows us to focus on the product that showed the greatest advantages in the different fluids. In a subsequent phase, it will be necessary to include the new MH_C product in a finished pharmaceutical form (tablet or capsule) and evaluate its technological characteristics and behavior in different biorelevant fluids such as FaSSIF and FeSSIF. Only after having collected all this information in vitro will it be possible to undertake an in vivo study to confirm the actual success of the approach considered. The obtained results are promising, as the MH_C sample reaches meloxicam dissolution higher than 90% in 1 h at pH 7.5, and in 5 h in H_2_O. The investigation can also be extended to other polymers in order to improve meloxicam solubility at lower pH values and increase the nanotube surface area.

## Data Availability

The data presented in this study are available on request from the corresponding author.

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
