# Peer review of "Design of Etched- and Functionalized-Halloysite/Meloxicam Hybrids: A Tool for Enhancing Drug Solubility and Dissolution Rate"

_pharmaceutics, 2024, doi:10.3390/pharmaceutics16030338_

Round 1

Reviewer 1 Report

Comments and Suggestions for Authors

Ms.Nr: pharmaceutics-2867649

Friuli V. et al.: Design of etched- and functionalized–halloysite/meloxicam hybrids: a tool for enhancing drug solubility and dissolution rate

Manuscript presents the synthesis of new meloxicam-halloysite and meloxicam-modified (etched and functionalized) halloysite composits to achieve better solubility/dissolution of the poorly soluble NSAID drug. The drug-clay systems are characterized impressively with wide range of solid phase and spectroscopic analytical methods. It was found that meloxicam (MEL) loading into HNT are high (about 40%), its structure and morphology have been retained, and weak interactions exist between the API and the carrier. The novelty of the study is obvious. Recent development of nanotechnology has opened new promising perspectives for the improvement of bioavailability of BCSII drugs. One of them is the usage of HNTs as carrier system. The preparation and structural and morphological characterization of the new composits are nicely done, well presented and the interpretation of the results are correct.

However, the aim of these formulations was to improve the dissolution rate of MEL, so these results would show the success of the project. Unfortunately, this part is the weak point of the manuscript. Only pharmacopoeial methods and no real biorelevant media were used to study the dissolution. Behaviour in FaSSIF, FeSSIF and other widely used biorelevant media would predict the performance of the hybrids in the human body. Further on, deionized water is not a biorelevant fluid and can not simulate a glass of tap water. The equilibrium solubility of the drug from the formulations was also investigated in deionized water, which is not a well defined medium for this, and it is not clear why different values were obtained. After dissolution from the composit, MEL must have one constant equilibrium solubility value. The large SD (~ 70%) shows that the presence of the halloysite matrix can disturb the solubility measurement. Authors have to address this question. I suggest the acceptance for publication after revision.

Critical remarks:

1. M is used for mole thus MEL should be used for meloxicam

2. The description of methods is not enough detail, references are missing (e.g. for shake-flask method)

3. Some abbreviations are not disclosed: EDS, TEM, BET

4. The solubility of MEL is pH dependent, so the pKa values have to be inserted to the text

5. Authors have to interpret why the SD of dissolution results in water and pH 7.5 buffer is so high (Fig. 10 and 11.).

6. What can be the reason that dissolution in pH 7.5 is almost 100%, why in water only ~ 25%. The pH difference between the two solvents itself does not explain this.

7.  MH_C was concluded as the best composit.  Considering oral application, its properties are suitable to make tablets?

8. Figure 1 is overcrowded with 6 lines, improve the quality to cut into two parts.

9. Some edition mistakes:

            - lines 27-28: the sentence is not understandable, has to be corrected

            - line 513: correct to: “functionalized HNT”

            - everywhere in the text: instead of “pristine” use “raw, original, unchanged” words

Comments on the Quality of English Language

see above

Author Response

Comments and Suggestions for Authors

Ms.Nr: pharmaceutics-2867649

Friuli V. et al.: “Design of etched- and functionalized–halloysite/meloxicam hybrids: a tool for enhancing drug solubility and dissolution rate”

Manuscript presents the synthesis of new meloxicam-halloysite and meloxicam-modified (etched and functionalized) halloysite composits to achieve better solubility/dissolution of the poorly soluble NSAID drug. The drug-clay systems are characterized impressively with wide range of solid phase and spectroscopic analytical methods. It was found that meloxicam (MEL) loading into HNT are high (about 40%), its structure and morphology have been retained, and weak interactions exist between the API and the carrier. The novelty of the study is obvious. Recent development of nanotechnology has opened new promising perspectives for the improvement of bioavailability of BCSII drugs. One of them is the usage of HNTs as carrier system. The preparation and structural and morphological characterization of the new composites are nicely done, well presented and the interpretation of the results are correct.

However, the aim of these formulations was to improve the dissolution rate of MEL, so these results would show the success of the project. Unfortunately, this part is the weak point of the manuscript. Only pharmacopoeial methods and no real biorelevant media were used to study the dissolution. Behaviour in FaSSIF, FeSSIF and other widely used biorelevant media would predict the performance of the hybrids in the human body. Further on, deionized water is not a biorelevant fluid and can not simulate a glass of tap water.

I understand very well the suggestion to use biorelevant fluids such as FaSSIF and FeSSIF, however in this preliminary phase we aimed to make the drug available as soon as possible after oral administration and this involves testing the product in fluids capable of simulating the conditions gastric (both fasting and full stomach) rather than intestinal ones such as FaSSIF and FeSSIF, so that at least a relevant fraction of the dose could be immediately available even in these conditions. In a second phase, we plan to test the finished pharmaceutical form also in other fluids or possibly also with a pH change test. About this, a sentence has been added in the conclusions. Furthermore, since the solubility of the drug decreases with decreasing pH, therefore gastric conditions are more challenging than intestinal ones.

The USP test was used anyway because it is an essential reference for all products containing oral meloxicam. At the same time, deionized water is very useful for understanding the variability of behavior of the compounds under examination in unbuffered conditions, when the very presence of the MEL (and possibly the carrier) could modify the pH of the medium, as was actually detected during the dissolution. Moreover, administration with a glass of water could also dilute the buffered conditions of the gastric environment by modifying the pH conditions.

This sentence is added to the text:

The dissolution tests were carried out in two fluids simulating the gastric environment (in fed and fasted conditions) because the drug first encounters these conditions in vivo and also because the drug has a pH-dependent solubility, less soluble in acidic and neutral environments and more soluble in a basic environment, then, these conditions are the most selective to highlight the differences in the behavior of the proposed compounds. The USP test was used anyway because it is an essential reference for all products containing oral meloxicam. At the same time, deionized water is very useful for understanding the variability of the behavior of the hybrids under examination in unbuffered conditions, when the very presence of the MEL (and possibly the carrier) could also significantly modify the pH of the medium, as detected during the dissolution test.

Deionized water was used to have a reference in an unbuffered fluid. Administration with a glass of water could dilute the buffered conditions of the gastric environment, modifying the pH conditions.

 The equilibrium solubility of the drug from the formulations was also investigated in deionized water, which is not a well defined medium for this, and it is not clear why different values were obtained. After dissolution from the composit, MEL must have one constant equilibrium solubility value. The large SD (~ 70%) shows that the presence of the halloysite matrix can disturb the solubility measurement. Authors have to address this question. I suggest the acceptance for publication after revision.

Instead, the solubility test was done in deionized water because it allows us to highlight the behavior of the MEL in unbuffered conditions (in this case the result is a value, not a process). Also, in this case, is not easy to get standardized conditions because different quantities of drug in solution modify in different ways the pH (and in this case saturation is reached with different MEL amounts). Furthermore, if we had used a buffer, we would have obtained different solubility values of the MEL for each different pH value of the buffer, precisely because it has pH-dependent solubility.

Critical remarks:

  1. M is used for mole thus MEL should be used for meloxicam

Done

  1. The description of methods is not enough detail, references are missing (e.g. for shake-flask method)

The following reference is added.

Veseli, A., Žakelj, S., Kristl, A. (2019). A review of methods for solubility determination in biopharmaceutical drug characterization. Drug development and industrial pharmacy, 45(11), 1717-1724.

  1. Some abbreviations are not disclosed: EDS, TEM, BET

Done

  1. 4. The solubility of MEL is pH dependent, so the pKa values have to be inserted to the text

Thank for the suggestion, the pKa value is added with the following reference.

Line 218: (Meloxicam pka = 4.08.

O'Neil, M.J. (ed.). The Merck Index - An Encyclopedia of Chemicals, Drugs, and Biologicals. Whitehouse Station, NJ: Merck and Co., Inc., 2006, p. 1006.

  1. Authors have to interpret why the SD of dissolution results in water and pH 7.5 buffer is so high (Fig. 10 and 11.).

As highlighted by the contact angle measurements (see below), wettability plays an important role in the dissolution phase of MEL and this factor could explain the high SD of some of the dissolution profiles of the analyzed compounds, in particular those characterized by an intermediate speed (not too fast and not too slow), such as those recorded in water and in phosphate buffer at pH 7.5. In these fluids, the powdered product initially remains on the fluid surface, then progressively gets wet, and then slowly sinks into the dissolution medium. In the other fluids, it tends to remain on the surface for the entire duration of the test.

A sentence has also been added in the text to explain the variability of these curves.

  1. What can be the reason that dissolution in pH 7.5 is almost 100%, why in water only ~ 25%. The pH difference between the two solvents itself does not explain this.

The solubility of MEL is highly pH dependent, in fact in unbuffered water the dose of the drug lowers the pH of the medium from 6.9 to 6.3, a value measured at the end of the dissolution test. While in other buffered fluids, the final pH does not change.

This sentence was added to the text.

7.MH_C was concluded as the best composit. Considering oral application, its properties are suitable to make tablets?

Yes, the compound could be suitable for oral application, in fact the next development of the research involves formulating it in tablets or capsules to evaluate a possible industrial production.

A sentence has been added in the conclusions as a possible future development of the project as requested also by the second reviewer.

The preliminary evaluation of the possible modifications of the halloysite carrier to improve the dissolution performance of MEL allows us to focus on the product that showed the greatest advantages in the different fluids. In a subsequent phase, it will be necessary to include the new MH_C product in a finished pharmaceutical form (tablet or capsule) and evaluate their technological characteristics and behavior in different biorelevant fluids such as FaSSIF and FeSSIF. Only after having collected all this information in vitro, it will be possible to undertake an in vivo study to confirm the actual success of the approach considered.

  1. Figure 1 is overcrowded with 6 lines, improve the quality to cut into two parts.

Done.

  1. Some edition mistakes:

- lines 27-28: the sentence is not understandable, has to be corrected

Thank for the suggestion. The sentence has been rephrased:

The meloxicam-halloysite composite, functionalized with chitosan, showed the best performance both in water and in buffer at pH 7.5: the drug is completely released in 4-5 hours in water and, in less than 1 hour in phosphate buffer.

- line 513: correct to: “functionalized HNT”

Done

- everywhere in the text: instead of “pristine” use “raw, original, unchanged” words

Done

Reviewer 2 Report

Comments and Suggestions for Authors

The manuscript presents a study on enhancing the solubility and dissolution rate of meloxicam, a poorly water-soluble drug. The research focuses on synthesizing and characterizing meloxicam-halloysite nanotube (HNT) composites. Various modifications to halloysite, such as acidic and alkaline etching or functionalization with APTES and chitosan, were explored. The study demonstrates that these drug-clay hybrids, especially the chitosan-functionalized halloysite composite, significantly enhance the dissolution rate of meloxicam compared to its unmodified form. The improved solubility and dissolution rate are attributed to the increased specific surface area and altered wettability due to the modifications. The study concludes that these composites, particularly the chitosan-functionalized variant, hold promise for improving the bioavailability of poorly soluble drugs like meloxicam. Overall, this is an interesting manuscript to be further considered. Please consider the following comments in revision.

1.    To enhance the repeatability and readability of the method section, it is important to detail each step in the synthesis of meloxicam-HNT composites. The catlog number and vendors of each reagent should be described.

2.    The current manuscript barely performed quantitative and statistical analyses. It will be crucial to quantitatively analyze the data from XRPD, FTIR, SEM and TEM. If available, include the data from different biological replicates and perform T tests.

3.    The manuscript will benefit if the potential mechanism of their findings can be somehow explained. For example, explain the physicochemical interactions between meloxicam and HNTs, focusing on how each modification aids solubility enhancement. Cite relevant literature or provide theoretical/computational/experimental results.

4.    The conclusion section is short. I suggest including more discussions regarding outlooks/limitations of this study.

Author Response

Comments and Suggestions for Authors

The manuscript presents a study on enhancing the solubility and dissolution rate of meloxicam, a poorly water-soluble drug. The research focuses on synthesizing and characterizing meloxicam-halloysite nanotube (HNT) composites. Various modifications to halloysite, such as acidic and alkaline etching or functionalization with APTES and chitosan, were explored. The study demonstrates that these drug-clay hybrids, especially the chitosan-functionalized halloysite composite, significantly enhance the dissolution rate of meloxicam compared to its unmodified form. The improved solubility and dissolution rate are attributed to the increased specific surface area and altered wettability due to the modifications. The study concludes that these composites, particularly the chitosan-functionalized variant, hold promise for improving the bioavailability of poorly soluble drugs like meloxicam. Overall, this is an interesting manuscript to be further considered. Please consider the following comments in revision.

1.To enhance the repeatability and readability of the method section, it is important to detail each step in the synthesis of meloxicam-HNT composites. The catlog number and vendors of each reagent should be described.

Thank you for the suggestion. The details have been added in the synthesis procedure, and catlog are given.

  1. The current manuscript barely performed quantitative and statistical analyses. It will be crucial to quantitatively analyze the data from XRPD, FTIR, SEM and TEM. If available, include the data from different biological replicates and perform T tests.

In the present paper the reported techniques have been applied to qualitatively demonstrate the effective loading of the meloxicam. The meloxicam quantification is instead carried out by UV spectroscopy and by EDS analysis (measurements on different spots). Standard deviations are added in Table S3. As concerns XRPD, we obtained the crystallite size by applying the Scherrer equation: the Full Width at Half Maximum (FWHM) value, used to calculate the crystallite size, is evaluated by peak profile fitting. Standard deviations are added in Table S1. From TEM analysis, we put into evidence possible roughness of the external nanotube surface and, in the case of alkaline etching, we reported in the paper:

“Alkaline etching (Errore. L'origine riferimento non è stata trovata.b) tends to thin nanotube walls and to open the lumen”.

We have further analyzed the TEM images of H, HCl- and NaOH- etched samples, by measuring the external and internal diameter of some nanotubes and calculating their ratio. Here we show some TEM images: we demonstrate the diameters ratio for H and acidic etching samples is higher than 3.3, while that of the alkaline etched one is lower than 3. These values are reported in the revised version of the paper.

H

HCl-etched

NaOH-etched

We're not sure we understand what the biological replicas are referring to. If the reviewer refers to dissolution tests, these are profiles and cannot be evaluated by applying conventional statistical tests.

3.The manuscript will benefit if the potential mechanism of their findings can be somehow explained. For example, explain the physicochemical interactions between meloxicam and HNTs, focusing on how each modification aids solubility enhancement. Cite relevant literature or provide theoretical/computational/experimental results.

The referee is right in giving us the suggestion. However, due to weakness of the drug-clay interactions, detected only in the functionalized samples by FT-IR analysis, it is not trivial to deeply investigate relationship occurring between the modification and drug dissolution behaviour. As reported in the paper, the best dissolution performances are obtained for the MH_C sample, and the benefits are related to the enhanced surface area of the support and the weak chitosan-meloxicam interactions. Some papers report the characterization of chitosan meloxicam composites, but the nature of the interaction (indeed weak) occurring between the polymer and the drug, is not discussed.

4.The conclusion section is short. I suggest including more discussions regarding outlooks/limitations of this study.

Thanks for the suggestion, the following sentences have been added in the conclusions section.

The preliminary evaluation of the possible modifications of the halloysite carrier to improve the dissolution performance of MEL allows us to focus on the product that showed the greatest advantages in the different fluids. In a subsequent phase, it will be necessary to include the new MH_C product in a finished pharmaceutical form (tablet or capsule) and evaluate its technological characteristics and behavior in different biorelevant fluids such as FaSSIF and FeSSIF. Only after having collected all this information in vitro, it will be possible to undertake an in vivo study to confirm the actual success of the approach considered. The obtained results are promising, as the MH_C sample reaches meloxicam dissolution higher than 90% in 1 h at pH 7.5, and in 5 h in H2O. The investigation can be extended to other polymers, in order to improve meloxicam solubility also at lower pH values and increase the nanotubes surface area.

Round 2

Reviewer 1 Report

Comments and Suggestions for Authors

I agree with the corrections, and accept the replies to my questions.